# Application of Finite Element Analysis in Modeling of Bionic Harrowing Discs

**DOI:** 10.3390/biomimetics4030061

**Published:** 2019-09-03

**Authors:** Benard Chirende, Jian Qiao Li, Wonder Vheremu

**Affiliations:** 1School of Agricultural Sciences, University of Mpumalanga, Private Bag X11283, Mbombela 1200, South Africa; 2Key Laboratory for Terrain-Machine Bionics Engineering (Ministry of Education), Jilin University, Changchun 130025, China

**Keywords:** finite element analysis, biomimetics, non-smooth surface, soil forces, harrowing disc, tillage

## Abstract

Ansys software was used to carry out three-dimensional finite element analysis (FEA) for biomimetic design of harrowing discs based on the body surface morphology of soil burrowing animals like dung beetle (*Dicranocara deschodt*) which have non-smooth units such as convex domes and concave dips. The main objective was to find out the effects of different biomimetic surface designs on reducing soil resistance hence the horizontal force acting on the harrowing disc during soil deformation was determined. In this FEA, soil deformation was based on the Drucker–Prager elastic–perfectly plastic model which was applied only at the lowest disc harrowing speed of 4.4 km/h which is within the limits of model. The material non-linearity of soil was addressed using an incremental technique and inside each step, the Newton–Raphson iteration method was utilized. The model results were analyzed and then summation of horizontal forces acting on the soil-disc interface was also done. An experiment was then conducted in an indoor soil bin to validate the FEA results. The FEA results are generally in agreement with those of the indoor experiment with a difference of less than or equal to the acceptable 10% with an average difference of 4%. Overall, convex bionic units gave the highest resistance reduction of 19.5% from 1526.87 N to 1228.38 N compared to concave bionic units.

## 1. Introduction

Tillage is a process where energy is applied to the soil in order to change its physical condition for the purpose of crop establishment in agriculture [1,2]. The energy depends on the tractor speed, depth of harrowing, tool and soil properties [3,4]. Most of the soil cutting tools in agriculture were developed after trial and error in the fields [1]. The disc harrow is a very important farm implement used for secondary tillage, particularly levelling the soil and breaking big clods into a fine tilth in order to improve germination percentage. Due to the free rotation of disc gang mounted on a disc shaft using bearings, the disc harrow can work well in hard soils and heavy trash conditions especially if the front disc gang has scalloped discs, and can also ride over stumps or obstacles in the soil easily. In addition, its power requirements are generally lower than those of disc ploughs, hence it is sometimes used for minimum tillage when field conditions are suitable. In order to further improve the performance of the disc harrow, biomimetics can play a pivotal role in reducing soil resistance. Bionics uses nature as an inspiration in the design through observing morphology and behavior of natural organisms. Through the process of evolution, natural organisms have experimented with form and function for at least three billion years and human beings can draw lessons from such organisms like dung beetles [5,6]. Drawing such lessons leads us to biomimetics, where structures, characters, elements, behavior, and interaction of the biological systems based on nature can be an inspiration for a new design concept and functional principle of a machine [7]. Three-dimensional finite element model (FEM) can then be used to optimize the design with minimum costs, and is very effective when results are validated experimentally [1,8,9].

### 1.1. Biomimetic Design

This paper focuses on improving tool characteristics through biomimetic design, the main thrust being the design and reduction of soil-tool contact area as inspired by the cuticle texture of soil burrowing animals such as ants and dung beetles. In order to carry out a biomimetic design, the functional, morphological and structural analyses of a living organism are first done and the viability for an implement is then considered based on the analyses [10,11].

For ants and dung beetles, their resistance to soil adhesion is an outcome of evolution and adaptation over billions of years [10,11]. All the information required for biomimetic design can be obtained through observing the sample of the living organism using a microscope and then parameterization leads to simplified details of the sample. Past studies have shown that the body surface morphologies of most soil burrowing animals have non-smooth structures such as convex domes, concave dips, steps and ridges which play important roles in reducing soil adhesion and friction during their movement [12,13]. Reducing soil adhesion and friction results in less soil resistance to deformation especially if the soil moisture content is between the liquid limit and plastic limit [12,13]. A lot of work on biomimetic design of implements like bulldozers and furrow openers has been done at Jilin University in China [13,14,15,16]. It was found that biomimetic non-smooth surface can reduce sliding resistance substantially by up to 23% and the convex domes were more superior to the concave dips in soil resistance reduction [13,14,15,16]. Extensive field research has been done at the aforementioned university on non-smooth surfaces. Even though Moayad applied FEA to a certain extent, his research was on a flat surface of a furrow opener and not a concave surface where the convex domes and convex dips are difficult to arrange in order to form effective non-smooth units [16]. Furthermore, it is more difficult to ensure that the nodes on FEM elements are aligned with the convex domes so that they are able to contact the soil element first, in order to properly simulate the actual disc harrowing.

According to the Coulomb equation, the cutting resistance of soil engaging implement is given by [9]:
(1)
F = Pc A + FN tan β

where Pc is the adhesion force between tool and soil (N/cm^2^), A the actual contact area (cm^2^), FN the normal force on the interface, and β the friction angle between the tool and the soil in degrees. From Equation (1), it is seen that the main factor affecting the cutting resistance is the contact area between the tool and soil, and this was key in choosing convex domes and concave dips as a way of trying to reduce the contact area A in Equation (1) through biomimetics. Chirende et al. have already done some indoor experiments on bionic disc ploughs where a significant reduction of 19% was recorded for bionic non-smooth surfaces with convex domes, however they did not apply finite element analysis in the study [17].

### 1.2. Finite Element Method

Many researchers have used the finite element method (FEM) to conduct their analyses of soil deformation. Some considered soil material as non-linear elastic, while others considered it as non-linear elastic–perfectly plastic [1,8,9,18]. For the latter, which is considered in this paper, the stress is directly proportional to strain until the yield point is reached. Beyond the yield point, the stress–strain curve is perfectly horizontal. The Drucker–Prager elastic–perfectly plastic model is one of the material models that has been adopted for conducting a FEM analysis of the soil deformation process [1,8,9]. This model was constructed originally to investigate and solve structural problems and was subsequently used to solve those in solid mechanics [8].

Several researchers have used the Drucker–Prager model to build up FEM models of soil cutting with simple tillage tools [1,8,9]. However, a few have applied three-dimensional FEM analysis of soil deformation coupled with biomimetic design of soil tillage implements with curved surface like disc harrow. Three-dimensional FEM modelling of biomimetic designs of furrow opener using FEM was done by Moayad, who concluded that FEM was very effective even though the results of the model and experiment differed by up to 20% for the horizontal force after verifying the results of the modeling process [16]. Furthermore, the furrow opener which he used is flat, making it easier to manipulate during modelling as compared to a harrowing disc used in this research which has a curved surface. Nidal also applied three-dimensional FEM analysis of soil deformation during disc ploughing but was based on the hyperbolic model developed by Duncan and Chang and not the Drucker–Prager elastic–perfectly plastic model used in this paper [1]. Furthermore, he only used plain discs without applying any biomimetics. The Drucker–Prager elastic–perfectly plastic model was applied by Tagar et al. who found that it gave better results compared to the Discrete Element Method (DEM) used in the past, but their cutting tool was just a furrow opener with a flat surface with no biomimetic modification [18].

## 2. Materials and Method

### 2.1. Determination of Parameters for Finite Element Modelling

In addition to the modulus of elasticity, Poisson’s ratio and soil metal friction which were obtained using both triaxial compression and direct shear tests, the soil cohesion, angle of internal friction and the angle of soil dilation were also determined by the triaxial compression test (Figure 1). The direct shear tests were only used for verifying results from the triaxial compression test for the former three parameters. All six parameters above are important in determining the behavior of soil during deformation under the Drucker–Prager model and the average results of five recordings are shown in Table 1. As determined by the oven dry method, the soil sample used for the triaxial tests had a moisture content of 23.25% (dry basis). The soil bulk density of 1620 kg/m^3^ was measured using the core method where a cylindrical core of the soil was removed from the soil bin and the soil was then also oven dried. The dry weight of the soil was divided by the volume of the cylinder to give the bulk density shown in Table 1. During the triaxial tests, a disc was pressed onto the soil surface under a pressure of 25 kPa for 15 s, then pulled off, and the normal adhesion forces were recorded [16]. In addition to the soil mechanical parameters recorded in Table 1, the 65Mn spring steel used to manufacture the harrowing discs had the Modulus of Elasticity and Poisson’s ratio as 2.1 × 10^11^ and 0.27 respectively according to the supplier catalogue.

### 2.2. Finite Element Method (FEM)

The main objective of using FEM is to model different kinds of bionic harrowing discs, and then carry out an analysis of performance of the different bionic disc designs. FEM simulation was carried out using ANSYS software to analyze both the conventional and the bionic harrowing discs. The 3-D models were drawn in a CAD system and then transferred to ANSYS in Sat format. The arrangement of bionic units on a disc surface using a CAD system are shown in Figure 2 and the density of convex domes and concave dips were varied as either 10% or 30%. To ensure that the area density of 10% and 30% was achieved, iterative calculations of the surface area covered by all convex domes or concave dips on the harrowing disc surface were performed, and the result was divided by the effective total surface area of the concave side of the harrowing disc. The effective total surface area was calculated by subtracting the area covered by the central hole from the total surface area of the concave side of the harrowing disc. The central hole is where where the the disc is fixed to the gang shaft. It has to be noted that the non-smooth structures were arranged in concentric circles in order to follow the soil path as the disc rotates [14,19]. This is one of the factors regarded as most important in designing non-smooth units and is described by Ren as soil motion tracks [14,19]. The other important factors are distribution of normal stresses and choice of non-smooth units. In addition, the enveloping surface of the non-smooth structures is considered as continuous and smooth. The CAD drawing models can still be modified in ANSYS software to meet the FEM requirements. This is done by selecting the defeaturing option which ensures compatibility between CAD and ANSYS.

Solid element with 8 nodes, 185 element type and Drucker–Prager model was used for element formulation and soil meshing as shown in Figure 3. For the harrowing disc, solid element with 10 nodes (45 element type) and elastic isotropic material model was used. The analysis was then conducted by adopting the surface–surface contacting (TARG 170 and CONTA 174). The solution criterion was selected as large displacement static.

For the boundary conditions, only three sides of the soil model were constrained in X, Y and Z directions respectively with the exception of the top face which was left unconstrained. Finally, the harrowing disc surface facing the soil model was tilted at an angle of 15° and then displaced in the Z direction by 20 mm, through incremental displacements of 5 mm at a time, monitoring the deformation process using the Newton–Raphson iteration technique.

### 2.3. Validation of FEM Results

After the finite element modelling exercise, an indoor experiment was carried out to validate the model, and make the necessary adjustments. A soil bin experiment was set up as follows: Soil was loosened using rotary tines, a scraper blade and a roller were then used for soil leveling and compaction, respectively. The soil used in the experiment had particle size distribution as shown in Table 2. After pretreatment to remove soluble salts from the soil as a precautionary measure, the particle size distribution was determined using sedimentation cylinders with internal depth of 340 ± 20 mm and a hydrometer for determining the effective depth after dispersion and shaking.

A total of 8 harrowing discs with biomimetic surfaces (convex domes and concave dips) were used in the experiment and one with plain conventional surface was the control. This makes a total of 9 discs which were used in both the finite element modelling and the indoor experiment.

#### 2.3.1. Preparation of Disc Samples

A total of 9 harrowing discs made from 65 Mn spring steel (1% manganese) were used in the indoor experiment in order to validate FEM results. The harrowing disc had 440 mm diameter, 60 mm concavity and a 5.5 mm thickness. The thickness and the concavity which are lower distinguish it from a ploughing disc, which tends to be more robust. Of the harrowing discs, 1 was just left plain, 4 had carved concave dips on the concave side of the harrowing disc and the last 4 had bionic ultra-high molecular weight polyethylene (UHMPE) convex domes fixed also on the concave side. UHMPE convex domes had a nipple at one end, which was then fixed into a hole drilled into the concave side of the harrowing disc using super glue in order to ensure firmness. Even though steel is much easier to weld, the UHMPE material was preferred because of its hydrophobicity [14,15]. As a precaution, no heat treatment was done on all the discs so as to avoid potential failure of material during biomimetic modification of the disc. A Universal Milling machine was used for making the concave dips and convex domes with a base diameter of 20 mm, heights of 1 mm and 3 mm and densities of 10% and 30% as shown Figure 4.

#### 2.3.2. Soil Bin Experiment

An electric carriage was used as a source of power to the disc harrowing unit fixed with a sensor for measuring the horizontal force (Figure 5a,b). The specification of the sensor showed an allowable horizontal force of 5 kN. Signals from the sensor were conducted using Wave Book/6 linked to Wave Book/512 for the acquisition of the signal [17]. The DASYLAB program on a computer was used to read the output of the experiment. To ensure accuracy, calibration of the sensor was done before and after the experiment by using known weights. A linear regression equation was then calculated showing correlation between the voltage output and the known weight force as shown in Figure 6. The experiment was done at a speed of 4.4 km/h and replicated three times. The fixed parameters on the harrowing disc was harrowing depth of 150 mm with a disc angle of 45° and tilted at an angle of 15°.

## 3. Results and Discussion

### 3.1. Von Mises Stress

Figure 7 shows the distribution of von Mises stress in the soil after disc displacement of 20 mm. For a plain disc in Figure 7a, the distribution is more even with more stress concentration at the center compared to the biomimetic discs. The concave geometrical nature of the harrowing disc largely influences the direction of passive cutting force acting on the soil. The resultant force tends to concentrate on the center of the disc. The effect of concave domes on the stress distribution is more pronounced in Figure 7i, where there are isolated spots of stress concentration, and these spots are on locations with convex domes more than on concave dips. The spots are a sign of reduced contact area between harrowing disc and soil, thus reducing the overall soil resistance through anti-friction and anti-adhesion. This is essentially the basis for achieving optimum biomimetic design with reduced soil resistance.

### 3.2. Derivation of Horizontal Force

To get the horizontal force on each node, the following path was used: ANSYS Main Menu–General Postproc–Nodal Calcs–Sum @ Each Node. Only the nodes on the soil disc interface were considered for summation of horizontal forces.

Due to the limitations of the Drucker–Prager elastic–perfectly plastic material model, the results are only compared at the lowest harrowing speeds of 4.4 km/h. Generally, the horizontal force results obtained from finite element methods were lower than those obtained from the indoor experiment (Table 3 and Figure 8). The coefficient of friction at the soil–plough interface could have been underestimated especially for bionic discs with UHMPE convex domes as it is generally very difficult to determine during the triaxial experiment. This is further supported by the plain disc which had the least difference of only 1.38% with more soil sticking on the disc surface compared to the bionic harrowing disc. It was difficult to correctly estimate the amount of soil which was going to stick on a bionic harrowing disc and factor it in the model. However, the difference between the model and the experiment was still less than 10%, in line with past FEM results [8,9,20]. Conforming to the general trend from past experiments, disc sample 9 gave the best performance with the highest draft force reduction of 15% compared to sample 1 which was just plain, and generally convex domes performed better than the plain ones and the concave dips [17,21]. The highest reduction of 15% is lower than that of the experiment which is at 19.5% reduction in horizontal force for sample 9 (Table 3 and Figure 8). Furthermore, convex units with bigger height of 3 mm were more effective in reducing the horizontal force most likely due to better ability to entrap more air between the non-smooth units and effectively reduce soil adhesion and friction. According to past researches, the biomimetic designed discs showed a significant reduction in drawbar power by up to 23% [15,16,17,21,22].

This finite element model used largely depends on the contact friction, geometry and the treatment of the soil–disc interface to get the best results. The FEM does not readily account for the effect of air trapped between the non-smooth structures and this may have affected the accuracy of the results as well.

## 4. Conclusions

The FEM method was applied at the lowest disc harrowing speed of 4.4 km/h and the results are generally in agreement with the indoor experiment results with an acceptable percentage difference of less than or equal to 10%, in line with past finite element models [8,9,20]. In both cases, the FEM and the indoor experiment showed that sample 9 was the most effective in reducing horizontal force and that the nature of bionic units (concave or convex) was most influential in reducing the resistance. The convex domes resulted in the highest horizontal force reduction.

According to the results, disc samples 1 to 9 showed a decreasing soil resistance in that order. Convex bionic surfaces gave highest resistance reduction reaching a maximum of 19.5% reduction for sample 9 in agreement with some past studies [15,16,17,21,22]. The increase in height of the bionic units resulted in reduced horizontal force. This is contradicting past researches by Ren et al. who found that there was no clear relationship between the bionic unit height and the reduction in horizontal force [15].

Finally, the bionic unit density of 30% which was highest also gave the highest resistance reduction compared to the 0% and 10% ones. This is in conformity with previously established results [15,16,17,21,22].

## Figures and Tables

**Figure 1 biomimetics-04-00061-f001:**
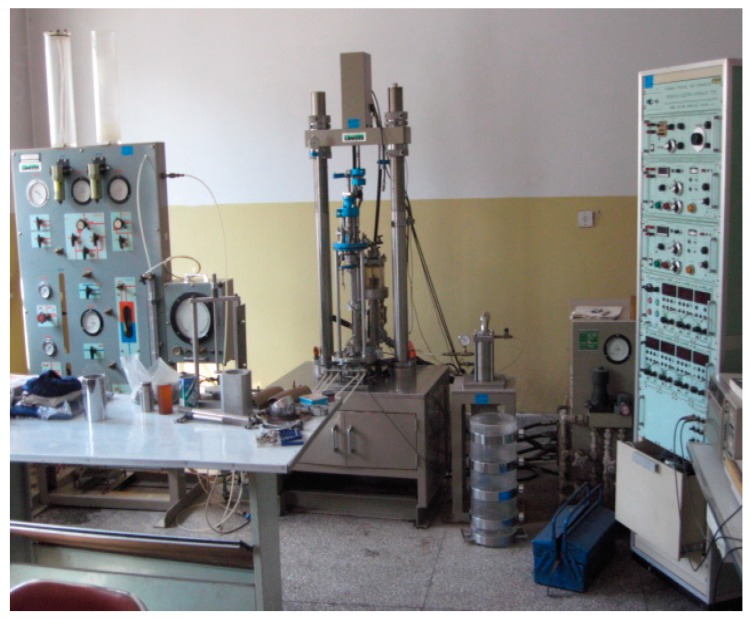
Triaxial equipment for measuring soil properties.

**Figure 2 biomimetics-04-00061-f002:**
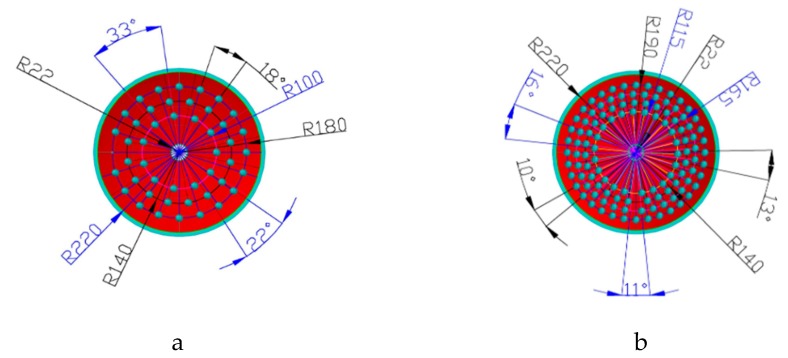
Arrangement of non-smooth units at 10% unit density (**a**) and at 30% unit density (**b**).

**Figure 3 biomimetics-04-00061-f003:**
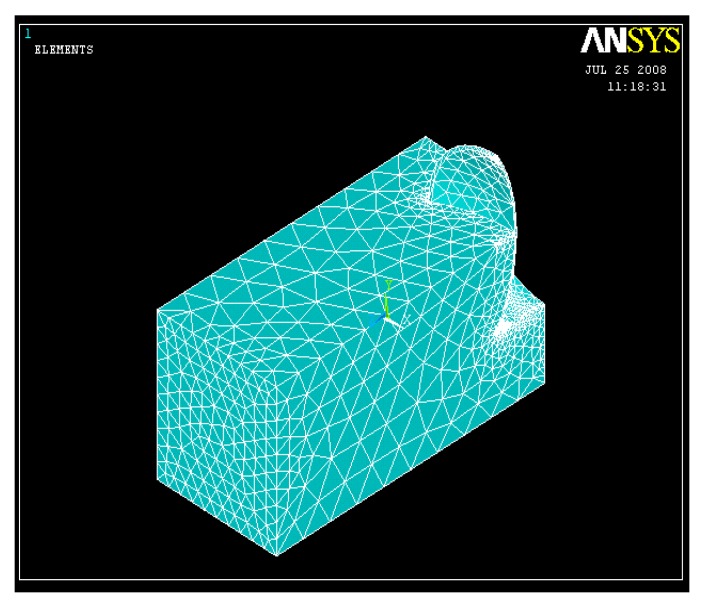
Mesh of the soil and harrow disc model (half symmetry).

**Figure 4 biomimetics-04-00061-f004:**
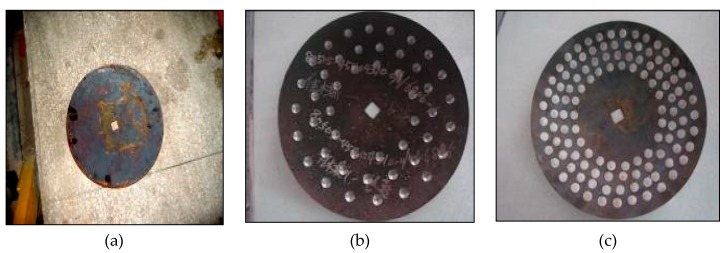
Bionic discs. (**a**) Plain disc; (**b**) bionic concave disc; (**c**) bionic convex disc.

**Figure 5 biomimetics-04-00061-f005:**
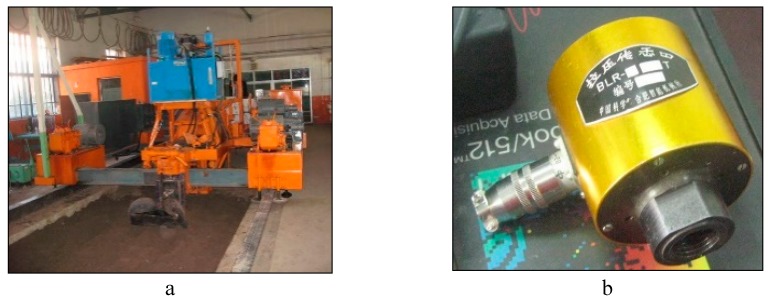
Indoor experiment equipment. (**a**) Indoor experiment electronic carriage; (**b**) 5 kN force sensor.

**Figure 6 biomimetics-04-00061-f006:**
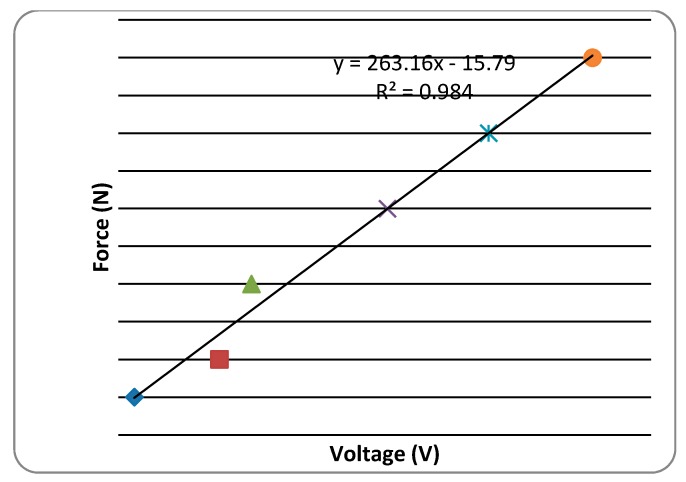
Calibration of force against horizontal voltage.

**Figure 7 biomimetics-04-00061-f007:**
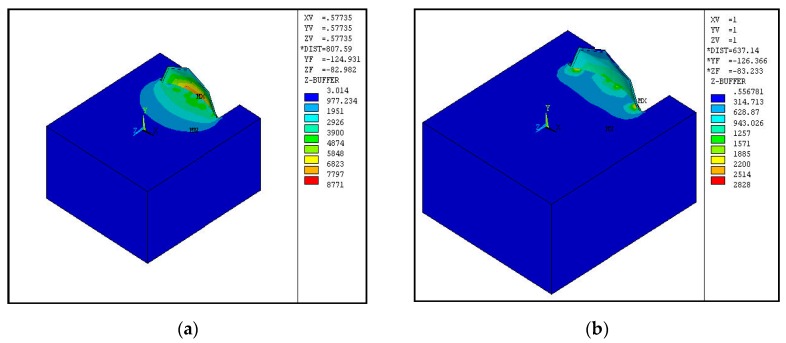
Von Mises Stress. (**a**) to (**i**): (**a**) Biomimetic disc (0% convex dome density which is basically the plain disc for control); (**b**) Biomimetic disc (1 mm deep, 10% concave dome density); (**c**) Biomimetic disc (3 mm, 30% concave hollow density); (**d**) Biomimetic disc (1 mm, 10% convex dome density); (**e**) Biomimetic disc (1 mm, 30% concave hollow density); (**f**) Biomimetic disc (3 mm, 10% concave hollow density); (**g**) Biomimetic disc (1 mm, 10% convex dome density); (**h**) Biomimetic disc (1 mm, 30% convex dome density); (**i**) Biomimetic disc (3 mm, 10% convex dome density).

**Figure 8 biomimetics-04-00061-f008:**
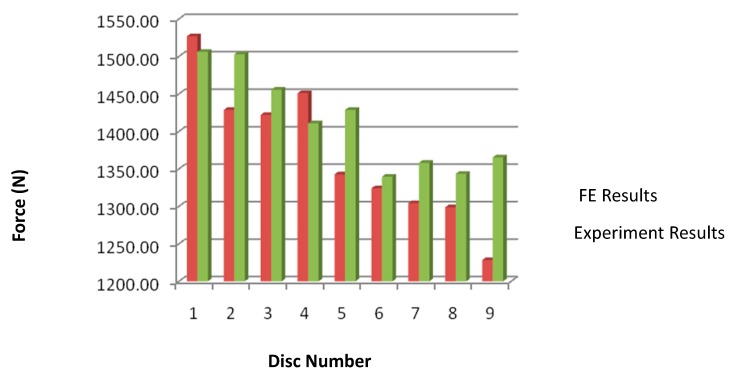
Comparison of finite element model with experimental results.

**Table 1 biomimetics-04-00061-t001:** Soil Mechanical Properties.

Soil Type	Bulk Density (kg/m^3^)	Cohesion (kPa)	Internal Friction (Degrees)	Soil Metal Friction (Degrees)	Angle of Dilation (Degrees)	Modulus of Elasticity (Pa)	Poisson’s Ratio
Black Loamy Soil	1620	10	34	23	4	8.067 × 10^6^	0.35

**Table 2 biomimetics-04-00061-t002:** Particle Size Distribution of the Tested Soil.

Particle Size (mm)	0.074–0.05	0.05–0.01	0.01–0.005	0.005–0.002
Weight (%)	33	42.5	15.5	9

**Table 3 biomimetics-04-00061-t003:** Comparison of finite element model with experimental results.

Disc Number	Disc Surface	Depth/Height	Density	FE Force (N)	Experiment Force (N)	(% Difference)
1	Plain	0 mm	0%	1526.87	1506.07	1.38
2	Concave	1 mm deep	10%	1428.67	1502.81	−4.93
3	Concave	1 mm deep	30%	1422.00	1455.63	−2.31
4	Concave	3 mm deep	10%	1450.92	1410.85	2.84
5	Concave	3 mm deep	30%	1342.65	1428.59	−6.02
6	Convex	1 mm high	10%	1324.05	1339.55	−1.16
7	Convex	1 mm high	30%	1304.21	1358.05	−3.96
8	Convex	3 mm high	10%	1298.78	1343.49	−3.33
9	Convex	3 mm high	30%	1228.38	1365.34	−10.03

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
