# Peer review of "Application of Finite Element Analysis in Modeling of Bionic Harrowing Discs"

_biomimetics, 2019, doi:10.3390/biomimetics4030061_

Round 1

Reviewer 1 Report

Most of the references to sources more than 10 years ago. It is recommended to add more recent research in this area.

Table 2 does not specify the dimension of the particle size parameter.

Section 2.3.1 says that 9 disks were made for the experiments (line 143). But then 10 disks are described (lines 146-147). Need to clarify.

Author Response

I have addressed the few comments from the reviewers as shown below:

The introduction has been improved as shown in red in the attached revised paper.

Most of the references to sources more than 10 years ago. It is recommended to add more recent research in this area (All added references from recent researches are shown in red in the reference list). Table 2 does not specify the dimension of the particle size parameter (The units is mm as shown in line 168). Section 2.3.1 says that 9 disks were made for the experiments (line 143). But then 10 disks are described (lines 146-147). Need to clarify. (The total number of discs used for modelling and the experiment were 9 (8 biomimetic modified ones and 1 plain disc acting as control) as corrected in line 174 to 176). 

Reviewer 2 Report

This manuscript focused on the resistance reduction effect of bionic Harrowing Discs by using FEA method. This work is meaningful and useful to the similar work in the future. The content of the manuscript is systematical, including simulations and the physical experimental verifications. However, some essential and critical concepts need to be made clear. 

General:

1.Line 17: The author mentioned “The material non-linearity of soil was addressed using an incremental technique and inside each step, the Newton-Raphson iteration method was utilized.” But I have not seen these method and technique again in material and methods which should refer to that. 

1.Line 57: The statement “non-smooth structures such as convex domes, concave dips, steps and ridges which play important roles in reducing adhesion and friction during their movement”, whereas, the aim of the manuscript is resistance reduction. I have doubts about the relationship between reducing adhesions and resistance reduction. Can the concave and convex used to reduce adhesion be used to reduce resistance? The author should clarify the relationship between the two above.

2.Line 61: “It was found that ... by up to 23 % and the convex domes were more superior to the concave dips in resistance reduction.” Does this statement have ref.? This statement seems confused with “19%” in Line76.

3.Line 102: "all the three parameters were measured using triaxial compression and direct shear test", but what about other soil parameters, such as bulk density, soil mental friction, modulus of elasticity and poisson’s ratio? How do you get them? As for the three parameters measured in the experiment in Table 1, are they the average values or the values measured once? 

4.Line116: “density of convex domes and concave…were varied between 10% and 30%”, how are these two density values determined? Please explain in details. The distribution of convex and concave in Figure 2 does not seem to be uniform, how is their position determined, or how is the degree of convex or concave spacing determined?

5.Line 140: How to get particle size distribution? It should be stated in the manuscript.

6.Line 165:"a disc angle of 45o and tilted at an angle of 15o." is mentioned in the physical experiment. Are these parameters consistent with the parameter settings in the simulation conditions? If consistent, it should be stated in the manuscript.

7.The Von Mises Stress of convex in Figure7 i, f is similar to concave in Figure 7b,e, no matter in the color distribution or in the value. Could the author give a clearer indication to assist the statement in the manuscript?

8.Line197: ”… sample 9 gave the highest draft force reduction of 15 % compared to the other samples”, “the other samples” here seems unclear.

9.Line219: “The increase in height of the bionic units resulted in reduced horizontal force.” This statement only appears in the conclusion, but not in the part of result and discussion. It should be added.

Specific:

1.Line 57 “non-smooth” is inconsistent with Line 61 “non smooth”, it should be unified. Line 80 “non-linear” and “non linear” are also need to be unified.

2.Some typographical errors appear in the text, such as unit symbols Line 71 ”(N/cm2)”, Line107 ” 2.1 x1011” and Line 165 ”45o”, which need to meet the specifications.

3.Figure 1 shows the chaotic experimental environment, and I think it would be better to change it.

4.The drawing label in Figure 2 does not meet the specifications.

5.Line135-140: Should this paragraph be placed in 2.3?

6.The arrangement of the subgraph in Figure 7 is confusing, and the title of the Figure 7 is wrong. These errors need to be correct.

Author Response

Please see the word attachment
